# Emerging Treatments Targeting the Tumor Microenvironment for Advanced Chondrosarcoma

**DOI:** 10.3390/cells13110977

**Published:** 2024-06-04

**Authors:** Vincenzo Ingangi, Annarosaria De Chiara, Gerardo Ferrara, Michele Gallo, Antonio Catapano, Flavio Fazioli, Gioconda Di Carluccio, Elisa Peranzoni, Ilaria Marigo, Maria Vincenza Carriero, Michele Minopoli

**Affiliations:** 1Preclinical Models of Tumor Progression Unit, Istituto Nazionale Tumori IRCCS ‘Fondazione G. Pascale’, 80131 Naples, Italy; vincenzo.ingangi@istitutotumori.na.it (V.I.); g.dicarluccio@istitutotumori.na.it (G.D.C.); m.minopoli@istitutotumori.na.it (M.M.); 2Histopathology Unit, Istituto Nazionale Tumori IRCCS ‘Fondazione G. Pascale’, 80131 Naples, Italy; a.dechiara@istitutotumori.na.it (A.D.C.); gerardo.ferrara@istitutotumori.na.it (G.F.); 3Musculoskeletal Surgery Unit, Istituto Nazionale Tumori IRCCS ‘Fondazione G. Pascale’, 80131 Naples, Italy; m.gallo@istitutotumori.na.it (M.G.); antonio.catapano@istitutotumori.na.it (A.C.); f.fazioli@istitutotumori.na.it (F.F.); 4Immunology and Molecular Oncology Diagnostics, Veneto Institute of Oncology IOV-IRCCS, 35128 Padua, Italy; elisa.peranzoni@iov.veneto.it (E.P.); ilaria.marigo@unipd.it (I.M.); 5Department of Surgery, Oncology and Gastroenterology, University of Padova, 35128 Padua, Italy

**Keywords:** chondrosarcoma, tumor microenvironment, angiogenesis, immune checkpoint inhibitors

## Abstract

Chondrosarcoma (ChS), a malignant cartilage-producing tumor, is the second most frequently diagnosed osseous sarcoma after osteosarcoma. It represents a very heterogeneous group of malignant chemo- and radiation-resistant neoplasms, accounting for approximately 20% of all bone sarcomas. The majority of ChS patients have a good prognosis after a complete surgical resection, as these tumors grow slowly and rarely metastasize. Conversely, patients with inoperable disease, due to the tumor location, size, or metastases, represent a great clinical challenge. Despite several genetic and epigenetic alterations that have been described in distinct ChS subtypes, very few therapeutic options are currently available for ChS patients. Therefore, new prognostic factors for tumor progression as well as new treatment options have to be explored, especially for patients with unresectable or metastatic disease. Recent studies have shown that a correlation between immune infiltrate composition, tumor aggressiveness, and survival does exist in ChS patients. In addition, the intra-tumor microvessel density has been proven to be associated with aggressive clinical behavior and a high metastatic potential in ChS. This review will provide an insight into the ChS microenvironment, since immunotherapy and antiangiogenic agents are emerging as interesting therapeutic options for ChS patients.

## 1. Introduction

### 1.1. Incidence and Histopathology

Chondrosarcoma (ChS) is the second most frequent malignant bone tumor, accounting for approximately 20% of all bone sarcomas [1]. These tumors constitute a heterogeneous group of mesenchymal neoplasms producing a non-osteoid cartilage matrix [2]. Epidemiologically, ChS seems to exhibit a weak predilection for males, with a mean age of incidence rate of 51 years [3]. Furthermore, rare ChS subtypes may also occur at a younger age, with a peak incidence in the second and third decades of life [4]. Recent data from the literature report that ChS rates are increasing, but this is probably due to the better characterization of the atypical cartilaginous tumor (ACT), which is a low-grade malignant ChS often localized in the appendicular skeleton [5,6]. 

The most common ChS localizations include bones of the axial skeleton, pelvis, scapula, sternum, and ribs, followed by the proximal humerus and femur. Of note, hands and feet are only rarely involved [7]. Macroscopically, ChSs are large tumors, usually greater than 4 cm in size, having a translucent lobular, white, or blue–grey cut surface due to the presence of hyaline cartilage [8]. 

Primary ChSs that arise de novo account for 20–27% of all primary malignant bone tumors, whereas secondary ChSs, accounting for approximately 1% of bone tumors, arise from pre-existing enchondroma or osteochondroma [9].

Grade evaluation of ChS is based on cellularity, nuclear features, and mitotic rate, as well as tumor matrix characteristics [10,11,12]. According to the 2020 WHO classification updated by Choi J. et al. [13,14], eight ChS subtypes have been identified. The conventional ChS is the most frequent subtype, accounting for about 25% of all primary bone tumors in adults. The majority of conventional ChSs are low- or intermediate-grade tumors characterized by an indolent clinical course and low metastatic potential. However, 5–10% of them exhibit high metastatic potential and a poor prognosis. The main site of metastatic disease is the lung, with regional lymph nodes less commonly involved [15]. According to their bone location, conventional ChS subtypes may be further classified as central (the most frequent) when they arise in the intramedullary cavity of bones, peripheral when they arise within the cartilage cap of a pre-existing osteochondroma, or periosteal (the rarest) when they localize on the surface of the bones. While the peripheral tumors are intermediate-grade or high-grade malignancies, the periosteal ChS seen on the external surface of the bone is Grade I or II. Cortical invasion, soft tissue extension, and size (>5 cm), as well as the presence of periostin, a stromal-related protein absent in enchondromas, can be helpful in distinguishing them from periosteal chondromas [16,17,18].

Dedifferentiated (dd) ChSs that occur in the fifth–seventh decades of life are rare, aggressive tumors accounting for approximately 10% of all ChSs [19]. Firstly described by Dahlin D.C. and Beabout J.W. in 1971 [20], ddChSs generally have a poor prognosis due to their strong tendency to metastasize. This histotype represents a great challenge for clinical management due to the coexistence of a high-grade non-cartilaginous sarcoma in the context of a low-grade chondrogenic component, with one blending of the first into the other [21]. The non-cartilaginous portion is frequently constituted by conventional osteosarcoma (70% of cases) and less frequently by malignant fibrous histiocytoma or fibrosarcoma [21].

Mesenchymal ChSs that frequently involve the axial skeleton are rare, high-grade tumors with a poor prognosis due to their strong tendency to metastasize. They can originate from either bone or soft tissues and are characterized by the presence of differentiated cartilage mixed with solid areas in which undifferentiated small round cells simulate Ewing’s sarcoma [9,22].

Finally, the very rare Clear Cell ChS is a low-grade malignant tumor occurring between the third and fifth decades of life. This subtype, which preferentially involves the epiphyses of long bones [23], is characterized by the presence of round or polygonal cells with large nuclei, centrally placed nucleoli, and abundant vacuolated, clear cytoplasm, arranged in lobules and sheets [16]. Woven bone formation and osteoclast-like giant cells are common features, but prominent osteoid formation should be considered for a differential diagnosis of osteosarcoma. Areas of conventional low-grade ChSs can be observed in approximately half of cases; mitotic figures are rare, but dedifferentiation can also occur [16].

### 1.2. Clinical Features and Therapeutic Management

The clinical presentation of ChSs often consists of persistent pain due to cartilaginous bone lesions with bone destruction, lytic, or blastic lesions with reactive thickening of the cortex or periosteum [24].

The prognosis of ChSs mainly depends on tumor grade and subtype. With respect to overall survival (OS), the ACT and secondary ChSs have a good survival rate, with a 5-year OS in more than 90% of cases [4,25]. Periosteal and conventional ChS subtypes have a good five-year OS (about 68%), followed by clear cell ChS (62.3%). One- and five-year OS in myxoid (64.1%, 49.8%) and mesenchymal (49.2%, 37.6%) subtypes are relatively lower but still better than in dedifferentiated ChSs, with reported five-year survival rates between 11% and 24% [26]. Survival decreases with increasing tumor grade, and the 5-year OS for grade 2 and grade 3 is 75% and 30%, respectively [4]. At initial presentation, only about 6% of ChS patients have a diagnosis of distant metastasis [7]. However, patients with metastatic disease at presentation have a 5-year OS of 28%, with a median OS of only 14 months [27]. Other negative prognostic factors include old age, tumor size, and tumor location in the axial skeleton [4].

At present, conventional radiation therapy is limited to patients with incomplete resection, inoperable tumors, or metastases [28], whereas systemic chemotherapy is only for the treatment of mesenchymal and ddChSs [29]. Nowadays, as discussed in a recent review study by Gilbert A. and colleagues [11], the introduction of hadron therapy seems to be a promising strategy for improving the local control and OS of ChS patients that are surgically inoperable. The chemo- and radio-resistance of ChS cells is mainly due to the presence of either a primary tumor mass or a metastatic niche of cancer stem cells (CSCs) that contributes to genetic and molecular heterogeneity [30]. This CSC subpopulation has been shown to play an important role in tumor progression, chemoresistance, recurrence, and metastasis [31,32]. Moreover, overexpression of the P-glycoprotein involved in the human multidrug resistance gene [33], a dense extracellular matrix, as well as poor vascularity that may establish a hypoxic microenvironment, contribute to the chemo- and radio-resistance of ChSs [34,35].

Nowadays, in addition to stem-like characteristics, it has been ascertained that genomic heterogeneity and epigenetic modifications in ChS cells, as well as deregulation of pivotal signal transduction pathways, are implicated in the drug- and radio-resistance of ChS cells [8,30,36]. This provided the rationale for developing new drugs, such as inhibitors of tyrosine kinase, activating mutated genes, mammalian target of rapamycin (mTOR), or histone deacetylase (HDAC) [37] (Figure 1). Unfortunately, most of them, including Hedgehog inhibitors that elicited good pre-clinical responses, failed to demonstrate a survival benefit when translated into clinical trials [38]. Although recent studies have demonstrated good short-term results after intralesional surgical procedures for low-grade ChSs treated with curettage and adjuvant treatments consisting of phenol application, cauterization, or cryotherapy, the primary and preferred treatment for ChS patients with localized disease remains radical surgical resection with wide margins [27,39,40,41]. Of note, for the management of extremity ChSs, amputation is only reserved for patients with extensive and invasive disease [42]. In this regard, the limited efficacy of available treatments combined with the clinical management of extensive and invasive ChSs remain challenging, and new therapeutic approaches are urgently needed [26].

In the last two decades, the role of the tumor microenvironment (TME) in governing the fate of solid tumors has been extensively investigated, and it is now clear that tumor cells establish an intricate crosstalk with cellular and soluble components within the TME, favoring tumor growth and progression [43]. In addition to tumor cells, the immune cells of the TME, depending on their nature and functional state, can exert cytotoxic activity, ensuring efficient immune surveillance or promoting tumor progression, making the TME immune-tolerant. The composition of the TME varies between tumor types, and hallmark features include the extracellular matrix (ECM), normal and transformed stromal cells, endothelial cells, and immune cells, as well as soluble components, including cytokines, chemokines, growth, and angiogenic factors (Figure 1) [44].

This review aims to provide insights into the components of the TME that govern ChS growth and progression, highlighting the mechanisms of action and clinical usage of antiangiogenic and immunotherapeutic agents.

## 2. Tumor Microenvironment

Although relatively few studies have allowed broadening the knowledge of the TME in ChSs, to date, it is clear that a plethora of signaling molecules within the TME cooperate to support ChS progression and metastasis through a variety of mechanisms [30].

### 2.1. Extracellular Matrix

The microenvironment of ChSs is characterized by a dense, heterogeneous ECM that provides a physical scaffold to support cellular functions and tissue integrity. In low-grade ChSs, the ECM is commonly organized in irregularly shaped and sized lobules separated by fibrous bands, and the occurrence of calcification areas arising from bone destruction suggests their enchondroma origin [5,12,21]. The ECM is a complex network in which several macromolecules, including collagen members, glycoproteins, and proteoglycans, as well as soluble factors, govern its continuous remodeling, playing important roles in angiogenesis, cancer invasion, and metastasis [45]. The ECM is continually remodeled by matrix metalloproteinases (MMPs) [46]. MMP-1, MMP-2, and MMP-13 regulate ECM degradation, and their high expression well correlates with ChS cell invasiveness and progression disease (PD), whereas MMP-9 expression has been documented to be associated with better survival in ChS patients [47,48,49].

Unlike normal chondrocytes that synthesize collagen types II, IX, X, XI, and proteoglycans [50], ChS cells produce a much more compact cartilaginous matrix, consisting of type II collagen, hydrophobic proteoglycans, and hyaluronan, a large negatively charged polysaccharide. In virtue of its tendency to aggregate, hyaluronan forms supramolecular complexes that contribute to the pericellular matrix organization [30]. By comparing hyaluronan content in ChS tissues with normal chondroid tissue as a control, Fede C. and colleagues found that a correlation exists between hyaluronan content, ChS aggressiveness, and chemoresistance [51]. Accordingly, the 4-methylumbellifer that is an inhibitor of hyaluronan synthesis prevents ECM formation, reducing in vitro and in vivo proliferation and invasion of rat ChS cells [52]. Recently, Nota S.P.F.T. and coworkers analyzed the expression levels of the Chondroitin Sulfate ProteoGlycan 4 (CSPG4) in a tissue microarray built with 29 primary conventional and 47 ddChS tissues by immunohistochemistry (IHC). The authors found high expression of CSPG4 in 71% of conventional ChS tissues and in 15% of ddChS tumors. In addition, CSPG4 expression was found to positively correlate with the time to metastasis and the survival of either conventional or ddChS patients [53]. The lysyl oxidase (LOX) family maintains the rigidity and structural stability of the ECM by catalyzing the cross-linking of collagen and elastin. Several studies have shown that aberrant expression of LOX in solid tumors, including ChSs, is associated with tumor progression and metastasis in virtue of the LOX’s ability to alter the ECM, increasing tumor cell migration and invasion [54,55].

Among the cytokines released in the ECM, Interleukin (IL)-1beta contributes to ChS vascularization and progression by inducing ChS cells to secrete the vascular endothelial growth factor (VEGF) [56]. Similarly, the CCL-5 chemokine (otherwise named RANTES) correlates with the tumor stage of ChS patients by inducing VEGF release [57]. All this evidence suggests that perturbation of the ECM is a possible strategy for inhibiting tumor growth and that deregulation of secretory trafficking of ECM molecules may provide viable therapeutic targets for the treatment of ChSs.

### 2.2. Hypoxia

The hypoxic environment that characterizes ChSs is mainly due to an imbalance occurring between oxygen delivery and consumption [58]. Low oxygen levels activate hypoxia-inducible factors (HIFs), which affect key stages of the ChSs’ metastatic process, including angiogenesis, invasion, tumor cell dormancy, release of extracellular vesicles, and progression of ChS metastases [59]. Low oxygen levels may increase HIF-1alpha expression in both normal and malignant chondrocytes. Analyzing 20 human ChS tissues, Kubo and colleagues found that both HIF-1alpha and HIF-2alpha are expressed in ChS cells and that HIF-1alpha content positively correlates with a shorter disease-free survival (DFS) [60]. Furthermore, a hypoxic microenvironment may induce chemo-resistance by allowing and selecting the survival of CSCs [61]. HIF-1 alpha and HIF-2 alpha factors trigger CSC proliferation via the activation of the PI3K/AKT and NF-κB pathways [62,63], allowing activation of Notch and Hedgehog pathways that maintain the stem potential of tumor cells [11]. Hypoxia also promotes cell invasion through HIF-1alpha, which increases MMP1 and activates the CXCR4/SDF-1 chemokine receptor axis in human ChS cell lines [64]. ChS vascularity increases with an increased histologic grade. This is due to the fact that, under hypoxic conditions, HIF-1alpha increases the release of VEGF, which, in turn, promotes tumor vascularization [65,66]. Kim H. and coworkers have documented that, like HIF-1alpha, HIF-2alpha upregulation is associated with a poor prognosis in high-grade ChS patients by conferring on ChS cells the capacity to grow, invade, and metastasize [67]. Furthermore, Hou S.-M. and coworkers recently found that, when cultured under hypoxic conditions, ChS cells secrete more exosomes that induce the polarization of macrophages into an M2-like phenotype, ultimately promoting a metastatic behavior in ChS tissues [68]. In the future, considering the pivotal role of hypoxia in sustaining the metastatic process, we foresee that the development of HIF-1alpha and/or HIF-2 alpha inhibitors could provide new therapeutic options for patients with metastatic ChS.

### 2.3. Adipose Tissue

The adipose tissue remodels the ECM through the secretion of MMPs, contributing to the adhesion and migration of ChS cells. Indeed, adiponectin levels released by differentiated adipocytes have been shown to correlate with the tumor stage of human ChSs. Lee HP and coworkers demonstrated that adiponectin induces ChS cells to secrete VEGF-A that, in turn, triggers migration and tube formation of human endothelial progenitor cells through PI3K, Akt, mTOR, and HIF-1alpha pathways [69]. Moreover, Chiu YC and colleagues found that adiponectin-dependent ChS cell migration is mediated by alpha2beta1 integrin upregulation [70]. The other adipocyte-derived cytokine, leptin, is increased in ChS tissues, and its content correlates with the ChS grade [71]. There is also evidence that leptin enhances angiogenesis and lymphangiogenesis in ChSs by promoting the secretion of VEGF-A and VEGF-C factors [72,73].

### 2.4. Angiogenesis

Although originating from avascular cartilaginous tissues, emerging data from the literature show that an intense microvessel density in ChS tissues is associated with aggressive clinical behavior, higher metastatic potential, and poor progression-free survival (PFS) [74]. Ayala G. and coworkers demonstrated that ChS vascularity increases with increased histologic grade and that blood vessels in ChSs may be located either in peri- or intra-cartilage [65]. Minopoli M. and coworkers investigated the relationship between clinical outcome and tumor vascularization in 18 ChS tissues, including 12 conventional and 6 ddChSs. They found that both ChS histotypes exhibit an appreciable intratumoral microvessel density, with vessels uniformly distributed in ddChSs or mainly localized at the margin of cartilaginous nodules in conventional ChS tissues. Data analysis supports the notion that an inverse correlation between high vascularization and shorter PFS occurs [75]. In addition to the direct and indirect contribution of the ECM and hypoxia in inducing angiogenesis, as already discussed above, other possible mechanisms are involved in ChS vascularization. For instance, Ingangi V. and coworkers demonstrated that peptide inhibitors of cell migration suppress neovascularization and intravasation of human ChS cells injected in nude mice [76,77], with a mechanism involving the urokinase receptor, formyl-peptide receptor type 1, and the integrin alphavbeta3 [78,79]. Another mechanism of angiogenesis in ChSs has been recently described by Cheng C. and colleagues. They found that exosomes derived from ChS cells are able to transport the long non-coding RNA (lncRNA) RAMP2-AS1 into human umbilical vein endothelial cells (HUVECs), promoting VEGFR2-dependent tube formation. Interestingly, the same authors also showed that elevated RAMP2-AS1 levels found in sera from ChS patients correlate with metastatic behavior and poor prognosis, suggesting that exosomal RAMP2-AS1 may be considered a novel biomarker and therapeutic target for ChS patients [80]. In addition to VEGFR, crucial mediators of ChS angiogenesis and metastasis are the platelet-derived growth factor (PDGF) receptors. A positive correlation has been observed between PDGFR-alpha and PDGFR-beta levels and ChS aggressiveness [81]. More recently, Wang CQ and coworkers documented that sphingosine-1-phosphate enhances expression of PDGF-alpha in human ChS cells, inducing angiogenesis through Ras, Raf, MEK, ERK, and AP-1 signaling pathways [82]. Once established the pivotal role of angiogenesis in affecting the PFS of ChS patients, efforts to develop antiangiogenic therapies have provided many agents, including small molecules, tyrosine kinase inhibitors, or fully human monoclonal antibodies such as bevacizumab that inhibit angiogenesis [83], and several of them have been introduced into clinical trials (Table 1). Firstly, imatinib, a multikinase inhibitor targeting PDGFR, was administered to 16 patients with unresectable or metastatic conventional ChSs without exerting meaningful clinical activity in terms of objective response (OR) and PFS lengthening [84]. Later, Phase 2 studies with other multikinase inhibitors, including pazopanib [85], regorafenib [86], and anlotinib [87], all targeting VEGF receptors, were found to improve the DFS of unresectable or metastatic conventional ChSs. Of note, the clinical trial with pazopanib in patients with unresectable or metastatic CS has completed accrual and is currently undergoing analysis for safety and efficacy. Finally, bevacizumab, a recombinant, humanized monoclonal antibody recognizing all four VEGF isoforms, has been included in several clinical trials since it provided beneficial effects when administered in combination with other drugs or as a maintenance regimen (Table 1). These recent results are very promising and will allow the inclusion of VEGF/VEGFR inhibitors in daily practice. Approaches based on the association of treatments are also in development, including clinical trials with chemotherapy and/or immune checkpoint inibitors (ICIs) associated with antiangiogenic agents (Table 1).

### 2.5. Immune Environment

Despite impressive advances in the understanding of how immune cells govern the progression of solid tumors [88], there are still few, sometimes contradictory, studies describing the immune landscape of ChSs. Nevertheless, increasing evidence seems to prove that a correlation between tumor tissue immunogenicity and the clinical course of the disease occurs in ChSs. In this regard, some authors reported that the immune infiltrate of ChS may include exhausted CD8+ T-cells [89] and that therefore some of these hard-to-treat patients could be treated with immunotherapeutic drugs. Moreover, given the importance of the localization of tumor-infiltrating lymphocytes (TILs) for response to immunotherapy in other solid tumors, it would be crucial to investigate the spatial distribution of these cells in ChS. Some observations exist for conventional (low-grade) tumors, where immune cells frequently localize at the periphery of the cartilaginous islands, probably due to the presence of a dense hyaline ECM that makes the tumor impenetrable [90,91]. Conversely, in some aggressive ddChSs with a more disorganized architecture, immune cells were also found in close proximity and intermingled with tumor cells [91], suggesting a potential responsiveness of these tumors to immunotherapy.

The TME is rich in immune infiltrates, including lymphocytes, neutrophils, natural killer cells, and macrophages. In ChSs, tumor-associated macrophages (TAMs) represent the most abundant immune cell population. TAMs mainly exhibit M2-like phenotypes that promote angiogenesis, extracellular matrix remodeling, cancer cell proliferation, metastasis, and immunosuppression, as well as resistance to chemotherapeutic agents [90,92,93]. Immunosuppression exerted by TAMs is mediated by the secretion of cytokines, chemokines, and MMPs that activate regulatory T-cells (Tregs) and block cytotoxic CD8+ T-cells through the secretion of NO and ROS metabolites [94]. Moreover, TAMs can recruit Tregs through CCL22, which further suppresses the antitumor immune response of T-cells [95]. It has been shown that a high number of CD68+/CD163+ M2-like TAMs positively correlate with metastatic diseases at diagnosis and shorter survival in sarcoma patients [90,92]. 

By IHC analysis of a large cohort of ChS tissues, including all ChS histotypes, Kostine M. and coworkers found that 85% of ddChSs are highly infiltrated by CD163/CD68-positive TAMs [93]. Accordingly, Simard FA and coauthors, using a swarm rat ChS model, showed that tumor size decreases and tumor progression delays upon either TAM depletion with clodronate-containing liposomes or their conversion into the pro-inflammatory M1 phenotype with mifamurtide [90]. Minopoli M. and coworkers demonstrated in multicellular 3D-organotypic models that human primary ChS cells induce monocyte-derived primary macrophages to acquire a M2-like phenotype, which, in turn, increases the invasive capability of ChS cells [75]. The authors also documented that, when human primary ChS cells were engrafted in nude mice, peptide inhibitors of cell migration reduced tumor size and vascularization by preventing recruitment and infiltration of murine monocytes into the tumor tissue [75]. A retrospective evaluation of a tissue microarray from 52 conventional and 24 ddChS tissues allowed Nota S.P.F.T. and coworkers to ascertain that an initially immunogenic environment may be reverted when ChS cells begin to utilize escape mechanisms that prevent their immune recognition and destruction [96]. TAMs can induce immunosuppression through the expression of the colony stimulating factor 1 receptor (CSF1R), also known as CD115. CSF1R may be engaged by two ligands, both of which are secreted by ChS cells: colony stimulating factor 1 (CSF-1) and interleukin-IL-34. Ligand-engaged CSF1R induces macrophages to release tumorigenic cytokines, including TGF-β and IL-10, which convert the immune microenvironment into a protumoral phenotype [97]. Richert J. and coauthors found that 84.6% of ddChSs exhibit a strong positivity for CSF1R macrophages, whereas in conventional ChSs it was only 64 % [91]. Importantly, CSF1R has been proposed to be an effective target for blocking TAMs that infiltrate the tumor stroma [98].

Several CSF1R inhibitors, including small molecules and neutralizing antibodies, have been developed, and many drugs targeting CSF1R are currently in clinical development or approved for treating solid tumors [99]. Therefore, agents blocking CSF1R could represent a novel therapeutic approach for ChS patients in the future.

Within the TME, Tregs have been shown to promote tumor progression by quenching antitumor immune responses. Furthermore, Treg may indirectly support the survival of cancer cells through their interaction with mesenchymal stromal cells [100].

In order to study the impact of TAMs and TILs on the clinical outcome of ChS patients, Richert J. and colleagues analyzed the phenotype and density of immune cells in 27 conventional and 49 ddChS tissues by IHC and RT-qPCR. The authors confirmed that TAMs are the major immune cell population in ChSs. Furthermore, they documented that the immune infiltrate composition in ddChSs correlates with patient outcome, as high levels of CD68+ macrophages are associated with the presence of metastases at diagnosis, whereas a high density of CD3+ TILs and CD8+ cytotoxic T-cells is associated with a better OS [91]. TAMs may lead to immunosuppression in sarcomas through the expression of several markers, including PD-L1, CD80, CD206, CD163, and CSF1R; the release of cytokines including IL-1, IL-6, IL-8, IL-10, TNF-alpha, TGF-beta, and VEGF; and chemokines such as CCL2, CCL5, CCL18, and MMP-9 [101]. Given the abundance of TAMs and the high level of expression of CSF1R reported, Richert J. and collaborators proposed that immunomodulation with CSF1R inhibitors may be a promising therapeutic approach in both conventional and ddChS ChS subtypes [91].

Discordant results have been reported from a multi-omic analysis of a large cohort of fresh ChS tumors. Based on the integrated data of single-cell CyTOF and flow cytometry, Li B. and coworkers provided the immune characterization of 98 conventional CHS cases. They classified the ChS immune microenvironment into three subtypes: the cluster “granulocytic–myeloid-derived suppressor cell (G-MDSC) dominant”, with a high number of myeloid cells; the immune exhausted cluster, with high exhausted T-cell and dendritic-cell infiltration; and the immune desert cluster, with few immune cells, the last subtype being frequently found in high-grade ChS. The results allowed authors to discourage immunotherapies targeting TAMs since few macrophages and monocytes were found in tumor tissues, while encouraging immunotherapeutic approaches with ICIs for ChS patients bearing the “immune exhausted” subtype [89].

In the TME, following TGF-β stimulation, neutrophils can be polarized toward N1 antitumor or N2 protumor phenotypes [102,103,104]. Like TAMs, tumor-associated neutrophils (TANs) impair antitumor immunity, stimulate angiogenesis, and accelerate tumor growth, invasion, and ultimately metastasis by remodeling the ECM and producing growth factors, including VEGF, a variety of cytokines, and chemokines such as IL-17 and MMP9, as well as the high mobility group box 1 [102,103,104]. Although many studies have evaluated the prognostic value of TANs as well as the neutrophil/lymphocyte ratio in a variety of solid tumors [105], to the best of our knowledge, only a few studies have been conducted in sarcoma, and none of them included ChS cases.

There is increasing evidence that the complement system is activated in various cancer tissues and has a role in the modulation of the tumor microenvironment. For instance, using U2-OS osteosarcoma cells in an in vitro model, Jeon H. and coworkers found that complement activation promotes angiogenesis by increasing the production of the proangiogenic factors FGF1 and VEGF-A [106], suggesting that a linkage between the complement system and angiogenesis exists and that the related mechanisms could be potential targets for ChS treatment.

C3a and its cognate receptor C3aR are key components of the complement system, having a role in tumor promotion and immunosuppression. C3a has been suggested to exert immunosuppressive activity by promoting neutrophil recruitment and CD4^+^ T-cell activation, as well as changing TAM polarization status and recruitment in a variety of solid tumors [107,108,109]. In this regard, Magrini E. et al. have recently ascertained that the C3a–C3aR axis plays a role in driving immunosuppression and tumor promotion in transplanted sarcoma models [109]. Therefore, the C3a–C3aR complex should be considered as a therapeutic target for counteracting the immunosuppressive activity of the tumor microenvironment.

## 3. Immunotherapy

The programmed cell death 1 (PD-1) receptor expressed on the surface of activated T-cells recognizes two ligands: PD-L1, otherwise named B7-H1, and PD-L2, otherwise named B7-DC. The interaction of these ligands with the immune checkpoint PD-1 on T-cells induces anergy in lymphocytes, counteracting T-cell-mediated immune responses [110]. When PD-L1 binds to PD-1, an inhibitory signal in T-cells blocks their proliferation and reduces cytokine production. Consequently, either the cytotoxic or the memory potential of CD8+ T-cells are impaired, resulting in the blocking of immune surveillance [111]. In recent years, blocking the PD-1/PD-L1 pathways with ICIs has changed the clinical management and fate of a variety of solid tumors [112]. The best-known ICIs are antibodies targeting PD-1, its ligand PD-L1, as well as the cytotoxic T-lymphocyte-associated protein 4 (CTLA-4). In this regard, the FDA, EMA, and other regulatory agencies approved several ICIs, including monoclonal antibodies targeting PD-1 (nivolumab pembrolizumab), PD-L1 (atezolizumab), and CTLA-4 (ipilimumab). Their impressive ability to elicit antitumor immune responses in solid tumors has encouraged investigations into their potential beneficial effects for ChS patients. Furthermore, aldesleukin, a recombinant form of human interleukin-2 that stimulates proliferation and maturation of the CD8-positive T- and natural killer-cells, has been shown to elicit promising outcomes in advanced solid tumors [113].

PD-1-positive T-cells have been found in both conventional and ddChS tissues [114]. Firstly, some years ago, Kostine M and coworkers, using a large tissue microarray from conventional, mesenchymal, and ddChS tissues, reported that PD-L1 is expressed in about 40% of ddChS tissues while it is absent in conventional and mesenchymal ChS tissues. The authors also found that PD-L1 levels in ddChSs are associated with the number of tumor-infiltrating leukocytes expressing the HLA-1 antigen [93]. Later, Yang X. and colleagues analyzed by IHC the expression levels of PD-L1 and PD-L2 in 40 human ChS tissues. Although the authors did not provide information on the analyzed histotypes, they found that PD-L1 and PD-L2 are expressed in 68% and 42% of ChS tissues, respectively, and that PD-L1 expression levels correlate with shorter survival [115]. Importantly, while PD-L2 expression levels only associate with earlier recurrence and lower age in ChS patients [115], PD-L1 expression has been found to be associated with larger tumor size, advanced tumor grade, earlier recurrence, and shorter OS in ChS patients [116]. Accordingly, a meta-analysis of 39 independent studies on 3680 patients with bone and soft tissue sarcomas, including 81 ChS cases, allowed Wang F. and coworkers to conclude that PD-L1 overexpression correlates with a higher rate of tumor metastasis, a more advanced tumor grade, and higher T lymphocyte infiltration [117].

These observations provided the rationale for testing the clinical efficacy of ICIs in counteracting ChS aggressiveness. Paoluzzi I. and coworkers reported in a retrospective study the first evidence of effects elicited by immunotherapy in ChS patients. The authors analyzed a cohort of 28 patients with relapsed, metastatic, or unresectable soft or bone sarcomas, including one mesenchymal ChS (primary site, skull) and two ddChSs (primary site, lower extremity of buttock). Patients received intravenous administration of 3 mg/kg nivolumab every 2 weeks. Two ChS patients had stable disease, whereas a partial response (PR) to nivolumab was achieved by a 74-year-old male with metastatic ddChS. Interestingly, higher PD-L1 levels in ChS cells from this patient were found, as compared with PD-L1 levels assessed in the tumor cells from the other ChS patients [118]. Subsequently, Tawbi H.A. and coauthors published the first results from the SARC028 clinical trial. This is the first multicenter, prospective Phase 2 study designed in order to examine the efficacy of pembrolizumab in bone and soft tissue sarcomas. Eighty-four patients with histological evidence of metastatic, surgically unresectable, or locally advanced sarcoma received an intravenous infusion of 200 mg pembrolizumab every 3 weeks until PD or unacceptable toxicity was ascertained. PD was assessed using computed tomography or magnetic resonance imaging at baseline, after 8 weeks, and then every 12 weeks until PD. Among 40 patients with bone sarcoma, 5 had ddChSs. For one of them, the response observed was satisfying, as a 50% tumor reduction and stable disease for more than 6 months were achieved. A second patient with ChS had stable disease, whereas the remaining three patients manifested progression disease, with a median follow-up of 18.7 months [119]. Furthermore, the correlative analyses of the SARC028 study documented that a higher density of cytotoxic tumor-infiltrating T-cells correlates with a better PFS [120]. Encouraging results were also derived from a Phase 1/2 nonrandomized clinical trial of doxorubicin plus pembrolizumab combined treatment in patients with advanced sarcoma, including one clear cell ChS, three conventional, and 4 ddChSs. As a result, three ChS patients had tumor regression, including one conventional ChS with a 26% decrease in size, and two ChS patients had durable disease regression, suggesting that doxorubicin plus pembrolizumab may provide benefit to these patients [121]. Further interesting evidence comes from a recent case report of a patient with femoral ddChS, multiple lung nodules, and a pelvic soft tissue mass. At the diagnosis, the authors reported PD-L1 expression in 95% of tumor cells and 20% in tumor-associated immune cells. The patient received 200 mg of pembrolizumab every 3 weeks. After five weeks, computed tomography showed a stable necrotic right pelvic lymph node and resolution of multiple lung nodules. After ten months, the patient reported the presence of an unchanged 4 mm pulmonary nodule and no metastatic disease in the abdomen and pelvis. Surprisingly, although after 29 cycles he discontinued the pembrolizumab treatment due to immunotherapy-induced psoriatic arthritis, the patient continued to have a complete response on surveillance imaging [122].

These aforementioned results evidence that some subtypes of ChSs may effectively benefit from treatment with immunotherapy, especially in the case of high-grade ChSs and ddChSs, as PD-1 inhibitors have been proven to stabilize the disease or elicit partial responses. Although responding patients seem to be associated with a higher number of tumor-infiltrating activated T-cells [89,116], it remains to definitively ascertain the potential predictive values of PD-L1 or PD-1 in ChSs. Based on these results, several clinical trials aimed at investigating the possibility that ChSs may effectively benefit from immunotherapeutic treatments, especially in the case of high-grade ones, have been planned, and most of them are ongoing (Table 1). However, it must be taken into account that most clinical studies include all sarcoma histotypes that should otherwise be considered distinct identities since sarcomas comprise more than a hundred histological subtypes and that, among these, ChSs comprise eight distinct histotypes. Therefore, our ability to select patients who will benefit from immune checkpoint blockade will remain poor until collaborative/multicenter studies allow us to enroll larger cohorts of ChS patients.

**Table 1 cells-13-00977-t001:** Recent, relevant, complete, and ongoing clinical trials with antiangiogenic and/or immune checkpoint inhibitors for ChS patients.

NCT Number	Clinical Trial	Phase	Drug Class	Drug	Patients (n. ChSs)	Outcomes	Status	Ref.
**NCT00928525**	Open-label trial of imatinib in patients with desmoid tumors and ChSs	2	Antiangiogenic MTKI	Imatinib– Mesylate	34 (16 ChSs)	Efficacy ORR	Completed	[84]
**NCT02982486**	Phase II of nivolumab plus ipilimumab in non-resectable sarcoma and endometrial carcinoma	2	Anti-PD-L1 Anti-CTLA-4	Nivolumab Ipilimumab	Soft and bone sarcomas	Efficacy Safety PFS OS	Unknown status	[123]
**NCT00464620**	Trial of dasatinib in advanced Sarcomas	2	Antiangiogenic MTKI	Dasatinib	109 (33 ChSs)	ORR CBR RR PFSR OS	Completed	[124]
**NCT02301039**	SARC028: A Phase II study of the anti-PD1 antibody pembrolizumab (MK-3475) in patients with advanced sarcomas	2	Anti-PD-1	Pembrolizumab	Soft tissue and bone sarcoma (n. 42 bone sarcomas)	OR CR PR PFS RR OS	Completed	[119]
**NCT03449108**	LN-145 or LN-145-S1 in treating patients with relapsed or refractory ovarian cancer, triple-negative breast cancer (TNBC), anaplastic thyroid cancer, osteosarcoma, or other bone and soft tissue sarcomas	2	Recombinant IL-2 BCL2 Inhibitor Anti-PD-1 Anti-CTLA-4 chemotherapy	Aldesleukin ALN-145 LN-145-S1 Cyclophosphamide Fludarabine Ipilimumab Nivolumab	30 (bone sarcoma, ddChS, giant cell tumor of bone, and 13 more)	Efficacy Safety ORR DCR PFS PR CR, SD DoR CR PR PD, PFS OS	Active, not recruiting	[125]
**NCT03474640**	Safety, tolerability, and pharmacokinetics of an anti-PD-1 monoclonal antibody in subjects with advanced malignancies	1/2	Humanized anti-PD-1	Toripalimab	Bone sarcoma	Safety Toxicity PFS PFSR OS	Closed for enrollment	[126]
**NCT01330966**	Study of pazopanib in the treatment of surgically unresectable or metastatic ChSs	1/2	Antiangiogenic MTKI	Pazopanib	47 ChSs	Safety Efficacy DCR PFS CR SD PD PR OS	Completed	[85]
**NCT03277924**	Trial of sunitinib and/or nivolumab plus chemotherapy in advanced soft tissue and bone sarcomas (ImmunoSarc)	1/2	Antiangiogenic MTKI Anti-PD-L1 chemotherapy	Sunitinib Nivolumab Epirubicin	200 (1 clear cell ChS, 6 dd-ChSs)	Safety Toxicity PFS OR CR PR OS	Recruiting	[127]
**NCT02888665**	Pembrolizumab and doxorubicin hydrochloride in treating patients with sarcomas that are metastatic or cannot be removed by surgery	1/2	Anti-PD-1 chemotherapy	Pembrolizumab Doxorubicin	Bone sarcomas (1 clear cell ChS, 3 Conventional, and 4 ddChSs)	ORR CR PR PFS MTD OS	Completed	[121]
**NCT02389244**	A Phase II study evaluating the efficacy and safety of regorafenib in patients with metastatic bone sarcomas (REGOBONE)	2	Antiangiogenic MTKI	Regorafenib	Metastatic bone sarcomas (46 ChSs)	PFS	Recruiting	[86]
**NCT05193188**	A multicenter clinically controlled study of anlotinib combined with PD-1 antibody on unresectable high-grade ChS with different IDH denotypes	2	Antiangiogenic MTKI PD-1 Ab	Anlotinib PD-1 Ab	high-grade, conventional, and ddChSs	PFSR ORR DCR	Recruiting	[87]
**NCT04055220**	Efficacy and safety of regorafenib as maintenance therapy after first-line treatment in patients with bone sarcomas (REGOSTA)	2	Antiangiogenic MTKI	Regorafenib	Bone sarcomas	Efficacy Safety RFS OS	Recruiting	[128]
**NCT04553692**	Phase I study of IGM-8444 as a single agent and in combination with subjects with relapsed and/or refractory solid cancers	1	Anti-DR5 Antiangiogenic chemotherapies BCL2 inhibitor	IGM-8444 Folfiri Bevacizumab Birinapant Venetoclax	Relapsed, refractory, or newly diagnosed solid cancers, including ChSs	ORR DoR PFS	Recruiting	[129,130]
**NCT03282344**	A study of NKTR-214 in combination with nivolumab in patients with metastatic and/or locally advanced sarcomas	2	IL2 pathway agonist, ICI	NKTR-214 Nivolumab	Metastatic and/or locally advanced soft and bone sarcoma	PR CR PD SD	Active, not recruiting	[131]
**NCT04458922**	Testing atezolizumab in people 2-17 years old with clear cell sarcoma or advanced ChS	2	Anti-PD-L1	Atezolizumab	27 ChSs	ORR CR PR, PD PFS RR OS	Active, not recruiting	[132]
**NCT03414229**	A study of epacadostat, an IDO1 inhibitor, in combination with pembrolizumab in patients with metastatic and/or locally advanced sarcomas	2	IDO1 Inhibitor Anti-PD-1	Epacadostat Pembrolizumab	Metastatic and/or locally advanced sarcoma	Efficacy Safety ORR PFS RFS OS	Active, not recruiting	[133]

Abbreviations: overall response rate (ORR), disease-free survival (DFS), clinical benefit rate (CBR), progression-free survival (PFS), progression-free survival rate (PFSR), disease control rate (DCR), duration of response (DoR), partial response (PR), complete response (CR), stable disease (SD), progressive disease (PD), response rate (RR), relapse-free survival (RFS), maximum tolerated dose (MTD), overall survival (OS).

## 4. Conclusions and Future Perspectives

Chondrosarcomas (ChSs) are a heterogeneous group of rare, cartilage-forming neoplasms, representing the second most common primary bone malignancy. Today, surgical management remains the main choice of treatments, as ChSs are resistant to both radiation and chemotherapy. Recently, the understanding of the immune landscape of ChSs has opened new immunotherapeutic options that will probably change the standard of care for high-grade ChSs and ddChSs in the future.

## Figures and Tables

**Figure 1 cells-13-00977-f001:**
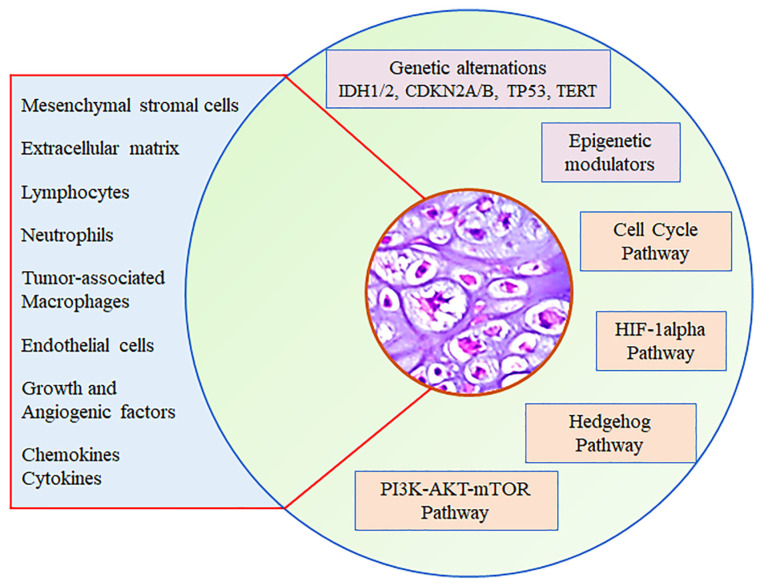
Schematic representation of multiple mechanisms by which genetic and epigenetic alterations, dysregulated signaling pathways, as well as soluble and cellular components within the TME, sustain the initiation and progression of chondrosarcomas.

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
