# Peer review of "Emerging Treatments Targeting the Tumor Microenvironment for Advanced Chondrosarcoma"

_cells, 2024, doi:10.3390/cells13110977_

Round 1
Reviewer 1 Report
Comments and Suggestions for Authors
The authors are to be commended on this thorough review. Just a few comments:
1. Would recommend against using disease progression (DP) and progressive disease (PD) as separate terms - should use PD as the generally accepted term
2. Similarly with survival rate (SR) and overall survival (OS) - stick with OS
3. Manuscript should be checked for typos
Comments on the Quality of English Language
Good
Author Response
Responses to the Reviewer #1:
The authors are to be commended on this thorough review. Just a few comments:
I sincerely thank you for your nice comments and suggestions
- Would recommend against using disease progression (DP) and progressive disease (PD) as separate terms - should use PD as the generally accepted term
- Similarly with survival rate (SR) and overall survival (OS) - stick with OS
- Manuscript should be checked for typos
1-2: I apologize for the mistake All DP and SR have be substituted with PD and OS, respectively and marked in red.
- . The entire manuscript has been reviewed by an author who worked in the United States. Now, all corrections are marked in red.
Reviewer 2 Report
Comments and Suggestions for Authors
This review on chondrosarcoma is well written and providing clinically relevant, timely topics.
Below are areas that have rooms for improvement.
1) From line 56 to 70, the authors concisely described chondrosarcoma subtypes. It would help readers' comprehension if the described features of subtypes are summarized in a table format.
2) Language editing will increase the quality. For example, capitalization is not required for common noun as seen in "the same Authors also..." on line 254. Another example on line 75 is a typo. "due the co-existence.." should be 'due to the ...'
3) Lines 99-101, "and mesenchymal (between 76.1% and 37.6%) subtypes have a relatively lower five-year survival as compared to the ddChS which retain a very low five-year SR (between11% and 24%) [26]. " This phrase is contradicting. It reads; 76.1 - 37.6% is lower as compared to 11 - 24%.
4) Line 262: "(PDGF)-alpha " Why parentheses are used? Wouldn't it be PDGFR-alpha? Or, do the authors mean PDGF-a, the ligand?
5) A brief explanation of biological significance of CD163/CD68 positivity (line 308) would be helpful.
6) The information given in the lines 428 - 429, "Further more, the correlative analyses... correlates with a better PFS", is disrupting the flow of presentation that describes the highlights of immune checkpoint inhibition data.
Comments on the Quality of English Language
Language editing will increase the quality. For example, capitalization is not required for common noun as seen in "the same Authors also..." on line 254. Another example on line 75 is a typo. "due the co-existence.." should be 'due to the ...'
Author Response
Responses to the Reviewer #2:
This review on chondrosarcoma is well written and providing clinically relevant, timely topics.
I sincerely thank you for your nice comments and suggestions
Below are areas that have rooms for improvement.
- From line 56 to 70, the authors concisely described chondrosarcoma subtypes. It would help readers' comprehension if the described features of subtypes are summarized in a table format.
Regarding the the histological classification of ChS histotypes, We realized that tables reporting the 2020 WHO classification have already reported by several authors, and are included in in several recently published reviews.
2) Language editing will increase the quality. For example, capitalization is not required for common noun as seen in "the same Authors also..." on line 254. Another example on line 75 is a typo. "due the co-existence.." should be 'due to the ...'
The entire manuscript has been reviewed by an author who worked in the United States. Now, all corrections are marked in red.
- Lines 99-101, "and mesenchymal (between 76.1% and 37.6%) subtypes have a relatively lower five-year survival as compared to the ddChS which retain a very low five-year SR (between11% and 24%) [26]. " This phrase is contradicting. It reads; 76.1 - 37.6% is lower as compared to 11 - 24%.
- We apologize for the mistake. The Paragraph on page 4 has been modified as follows: “Periosteal and conventional ChS subtypes have a good five-year OS (about 68%), followed by clear cellChS (62.3%). One- and five-year OS in myxoid (64.1%, 49.8%), and mesenchymal (49.2%, 37.6% subtypes are relatively lower but still better than in dedifferentiated CHS, with reported five-year survival rates between 11% and 24% [26].”
- Line 262: "(PDGF)-alpha " Why parentheses are used? Wouldn't it be PDGFR-alpha? Or, do the authors mean PDGF-a, the ligand?
Thanks for saving me from this mistake. We intend the PDGF Receptor alpha. Now the parentheses have been deleted.
- A brief explanation of biological significance of CD163/CD68 positivity (line 308) would be helpful.
This information has been added on page 8 as follows:” It has been shown that a high number of CD68+/CD163+ M2-lik TAM positively correlate with metastatic diseases at diagnosis and shorter survival of sarcoma patients [90,92]. By IHC analysis of a large cohort of ChS tissues, including all ChS histotypes, Kostine M. and coworkers found that 85% of ddChS are highly infiltrated by CD163/CD68 positive TAM [93].
- The information given in the lines 428 - 429, "Further more, the correlative analyses... correlates with a better PFS", is disrupting the flow of presentation that describes the highlights of immune checkpoint inhibition data.
The sentence “Furthermore, the correlative analyses of the SARC028 study documented that a higher density of cytotoxic tumor-infiltrating T cells correlates with a better PFS.
Reviewer 3 Report
Comments and Suggestions for Authors
The authors provide an overview over advanced chondrosarcoma treatment targeting the tumor environment.
The subtypes of the tumor entity were explained. The manuscript is well written and seems to be comprehensive. Figure 1: only some pathways are depicted. The manuscript reports also about Notch, HIValpha/beta, NFkappaB cascades e.g. lines 205-206 – should it be included?
More information concerning the tumor extracellular matrix might be helpful. Is it more hyaline or fibrocartilage-like containing collagen type I? ist he surrounding connective tissue periost-like?
Line 179: „Authors“ –, line 209, line 254, 393, 396, 409: „Axis“, do not write it with capital letter
The initials of the first author are not consistently included into the citations (compare lines 212 or 215)
Line 219: HIValpha/beta: could both inhibitors be combined?
2.3.
What does it mean – is the progress of the ChS more serious if adipose tissue is in the immediate environment? It means surrounding fatty tissue, there could also be e.g. fatty bone marrow in the bone metaphysis containing a lot of fat, however, these adipose cells derive from reticulum cells – do these cells have the same influence?
Line 229: alpha2beta1 integrins – could this receptor serve as a target?, line 250: alphaVbeta3 integrin-does it act in a similar manner?
Line 266: the point is surplus, similar in line 368
Line 268: insert a blank
Line 272: remove surplus bracket
Line 290: „TIL“ this abbreviation is explained later (line 338)
Line 294: „impenetrable“ how large ist he exclusion limit of the ECM
Line 313: „Organotypic“ do not use capital letter „They“ line: 342, 397
2.5. Immune Environment (could this paragraph be subdivided in subsections?)
Cellular immunity is explained – how about complement activity?
Line 390: insert blank
Line 411: write correctly „mesenchymal“
Line 449: it remains
Comments on the Quality of English Language
only minor inconsistencies as indicated
Author Response
Responses to the Reviewer #3:
The authors provide an overview over advanced chondrosarcoma treatment targeting the tumor environment. The subtypes of the tumor entity were explained. The manuscript is well written and seems to be comprehensive.
I sincerely thank you for your nice comments and suggestions
Figure 1: only some pathways are depicted. The manuscript reports also about Notch, HIValpha/beta, NFkappaB cascades e.g. lines 205-206 – should it be included?
Now, only the HIF-1alpha pathway has been included in the revised Figure 1. We did not include the others whose inhibitors have not yet been translated into the clinical practice for patients with ChS.
More information concerning the tumor extracellular matrix might be helpful. Is it more hyaline or fibrocartilage-like containing collagen type I? ist he surrounding connective tissue periost-like?
In the revised version, the following sentences have been included in the Introduction on page 3 as follows:” Cortical invasion, soft tissue extension, and size (> 5 cm) as well as the presence of periostin a stromal-related protein absent in enchondromas can be helpful in distinguishing them from the periosteal chondroma” ….and in the Extracellulr Matrix section on page 5 as follows: In low grade ChS, the ECM is commonly organized in irregular shaped and sized lobules separated by fibrous bands and the occurrence of calcification areas arising from bone destruction suggests their enchondroma origin [5,12,21].
Line 179: „Authors“ –, line 209, line 254, 393, 396, 409: „Axis“, do not write it with capital letter
The initials of the first author are not consistently included into the citations (compare lines 212 or 215) it has been corrected and maked in red.
Line 219: HIValpha/beta: could both inhibitors be combined?
What does it mean – is the progress of the ChS more serious if adipose tissue is in the immediate environment? It means surrounding fatty tissue, there could also be e.g. fatty bone marrow in the bone metaphysis containing a lot of fat, however, these adipose cells derive from reticulum cells – do these cells have the same influence? This is a very interesting question. However, unfortunately, I don't know the answer to this question.
Line 229: alpha2beta1 integrins – could this receptor serve as a target?, line 250: alphaVbeta3 integrin-does it act in a similar manner? Integrins play a pivotal role in governing cell ability to adhere, migrate and invade and thereby they may be considered as biomarkers. However, to the best of our knowledge, only few studies have been conducted with inhibitors of integrin functions in solid tumors and none of them included ChS cases.
Line 266: the point is surplus, similar in line 368 it has been corrected and maked in red.
Line 268: insert a blank it has been corrected and maked in red.
Line 272: remove surplus bracket it has been corrected and maked in red.
Line 290: „TIL“ this abbreviation is explained later (line 338) It has been corrected and maked in red.
Line 294: „impenetrable“ how large ist he exclusion limit of the ECM. Unfortunately, I don't know the answer to this question.
Line 313: „Organotypic“ do not use capital letter „They“ line: 342, 397 it has been corrected and marked in red.
2.5. Immune Environment (could this paragraph be subdivided in subsections?) We chose to use a single paragraph because the functions of immune cells are interconnected and, are described simultaneously in the cited articles.
Cellular immunity is explained – how about complement activity?
This is a very interesting question. In the revised Manuscript the following sentences have been included on pages 9-10: “There is increasing evidence that the complement system is activated in various cancer tissues having a role in the modulation of tumor microenvironment. For instance, using U2-OS osteosarcoma cells in an in vitro model, Jeon H. and coworkers found that the complement activation promotes angiogenesis by increasing the production of the proangiogenic factors FGF1 and VEGF-A [106], suggesting that a linkage between the complement system and angiogenesis exists and that the related mechanisms could be potential targets for ChS treatment.
The C3a and its cognate receptor C3aR are key components of the complement system having a role in tumor promotion and immunosuppression. C3a has been suggested to exert immunosuppressive activity by promoting neutrophil recruitment, CD4+ T-cell activation, as well as changing TAM polarization status and recruitment in a variety of solid tumors [107-109]. In this regard, Magrini E. et al, have recently ascertained that the C3a–C3aR axis have role in driving immunosuppression and tumor promotion in transplanted sarcoma models [109]. Therefore, the C3a/C3aR complex should be considered as a therapeutic target for counteracting the immunosuppressive activity of tumor microenvironment”.
Line 390: insert blank t has been corrected and maked in red.
Line 411: write correctly „mesenchymal“t has been corrected and marked in red.
Line 449: it remains t has been corrected.